# UK Dietary Practices for Tyrosinaemias: Time for Change

**DOI:** 10.3390/nu14245202

**Published:** 2022-12-07

**Authors:** Anne Daly, Sarah Adam, Heather Allen, Jane Ash, Clare Dale, Marjorie Dixon, Carolyn Dunlop, Charlotte Ellerton, Sharon Evans, Sarah Firman, Suzanne Ford, Francine Freedman, Joanna Gribben, Sara Howe, Farzana Khan, Joy McDonald, Nicola McStravick, Patty Nguyen, Natalia Oxley, Rachel Skeath, Emma Simpson, Allyson Terry, Alison Woodall, Lucy White, Anita MacDonald

**Affiliations:** 1Birmingham Women’s and Children’s Hospital, NHS Foundation Trust, Steelhouse Lane, Birmingham B4 6NH, UK; 2Royal Hospital for Children, Glasgow G51 4TF, UK; 3Sheffield Children’s NHS Foundation Trust, Sheffield S10 2TH, UK; 4University Hospital of Wales, Cardiff CF4 4XW, UK; 5University Hospitals Birmingham NHS Foundation Trust, Birmingham B15 2TH, UK; 6Great Ormond Street Hospital for Children NHS Foundation Trust, London WC1N 3JH, UK; 7Royal Hospital for Sick Children, Edinburgh EH9 1LF, UK; 8University College London Hospitals NHS Foundation Trust, London WC1N 3BG, UK; 9Guy’s and St Thomas’ NHS Foundation Trust, London SE1 7EU, UK; 10Southmead Hospital North Bristol Trust, Bristol BS10 5NB, UK; 11Evelina London Children’s Healthcare, London SE1 7EH, UK; 12Bradford Teaching Hospitals, NHS Foundation Trust, Bradford BD5 0NA, UK; 13Belfast Health and Social Care Trust, Belfast BT9 7AB, UK; 14Royal Manchester Children’s Hospital, Manchester M13 9WL, UK; 15Alder Hey Children’s NHS Foundation Trust, Liverpool L12 2AP, UK; 16Salford Royal NHS Foundation Trust, Manchester M6 8HD, UK

**Keywords:** tyrosinaemia type I, II, III, dietary treatment, protein, tyrosine, phenylalanine

## Abstract

In the UK, different dietary systems are used to calculate protein or tyrosine/phenylalanine intake in the dietary management of hereditary tyrosinaemia, HTI, II and III (HT), with no systematic evidence comparing the merits and inadequacies of each. This study aimed to examine the current UK dietary practices in all HTs and, using Delphi methodology, to reach consensus agreement about the best dietary management system. Over 12 months, five meetings were held with UK paediatric and adult dietitians working in inherited metabolic disorders (IMDs) managing HTs. Eleven statements on the dietary system for calculating protein or tyrosine/phenylalanine intake were discussed. Dietitians from 12 of 14 IMD centres caring for HT patients participated, and 7/11 statements were agreed with one Delphi round. Nine centres (three abstentions) supported a 1 g protein exchange system for all foods except fruit and vegetables. The same definitions used in the UK for phenylketonuria (PKU) were adopted to define when to calculate foods as part of a protein exchange system or permit them without measurement. Fruit and vegetables contain a lower amount of tyrosine/phenylalanine per 1 g of protein than animal and cereal foods. The correlation of tyrosine vs. phenylalanine (mg/100 g) for vegetables and fruits was high (r = 0.9). In Delphi round 2, agreement was reached to use the tyrosine/phenylalanine analyses of fruits/vegetables, for their allocation within the HT diet. This allowed larger portion sizes of measured fruits and vegetables and increased the variety of fruit and vegetables that could be eaten without measurement. In HTs, a combined dietary management system will be used: 1 g protein exchanges for cereal and milk protein sources and tyrosine/phenylalanine exchanges for fruit and vegetables. Intensive, systematic communication with IMD dietitians and reappraisal of the evidence has redefined and harmonised HT dietary practice across the UK.

## 1. Introduction

Hereditary tyrosinaemia types I, II and III (HT) belong to a group of rare autosomal recessive, inherited metabolic disorders (IMDs). HTI, the most clinically challenging of these disorders, is caused by mutations in the gene fumarylacetoacetate hydrolase (FAH) leading to the accumulation of toxic metabolites causing apoptosis and liver and renal failure [1,2]. HTII, tyrosine aminotransferase deficiency, causes oculocutaneous and hyperkeratosis of the feet and hands [3,4]. HTIII, the rarest of the HTs, has a variable neurological outcome that is inadequately defined [5,6]. Patients can present with symptoms at different ages as these conditions are not included in the UK newborn screening programme.

HTs are managed with a protein or tyrosine/phenylalanine-restricted diet. HTI is also managed with 2 (2-nitro-4-trifluoromethybenzoyl)-1, 3 cyclohexanedione) (NTBC), preventing the production of toxic metabolites by blocking the tyrosine pathway at step 3, so mimicking HTIII. For all HTs, the individual tolerance to protein or tyrosine/phenylalanine intake is assessed by the provision of natural protein-containing foods and regular monitoring of blood tyrosine and phenylalanine concentrations. In the UK, we aim to maintain a target tyrosine treatment reference range of 200–400 µmol/L and a phenylalanine concentration of ≥50 μmol/L [7]. The diet is supplemented with a free/low-tyrosine/phenylalanine protein substitute (based on L-amino acids or glycomacropeptide) and low-protein special foods, fruits and some vegetables. Dietary management of HTs is variable both within and between various countries [8,9] with no international standardisation due to an absence of international scientific guidelines. Additionally, the practical dietary application is divergent between the different HTs, commonly associated with the rarity, diversity and limited management experience of these disorders. Some HTIII patients do not require dietary restriction beyond early childhood and still achieve blood tyrosine concentrations within therapeutic treatment range [10,11]. There is no systematic evidence comparing the merits and inadequacies of the different dietary systems in use.

Several collaborative European, Canadian and USA clinical guidelines are published [9,12,13] by working groups on HTI but without consensus guidance on dietary management. Dietetic representation on any of these working groups was limited. Recommendations on blood tyrosine concentrations for HTI are variable ranging from 200 to 800 µmol/L [9,13], which has implications for the stringency of dietary management required. Evidence suggests blood phenylalanine concentrations should be >50 µmol/L, and although this is widely acknowledged, it is not universally adopted in clinical practice [7]. There is also inadequate data on long-term clinical outcomes, blood tyrosine/phenylalanine control and dietary protein/tyrosine tolerance in all conditions. Management for HTI changed in 1990 with the introduction of NTBC, meaning most patients with HTI on this therapy are <30 years old.

In the UK, commonly a 1 g protein exchange system (1 g of protein = 1 exchange) is adopted for allocation of foods such as milk and cereals, but practices vary for fruit and vegetables, with some centres using 1 g protein exchanges [14] and some applying the same dietary system used in patients with PKU (personal communication). There is limited, inconsistent and fragmented analysis about the tyrosine content of fruits and vegetables, and therefore, many IMD dietetic teams use a surrogate marker (i.e., their protein or phenylalanine content) to estimate intake. It is also unclear about the guidance to determine when foods should be calculated/measured as part of the protein exchange system or when the protein content is so low it can be considered an ‘exchange-free’ food so included without measurement.

Following a successful collaborative project by the British Inherited Metabolic Disease Dietitians Group (BIMDG-DG) in which consensus for the dietary management of PKU was achieved [15,16], the same collaborative Delphi methodology was used to gather consensus on the dietary management of all HTs. The views of physicians were not sought in this process as it was a dietetic based project. The aim was to reach a national approach for the dietary guidance of all HTs.

## 2. Methods

Over 12 months (January 2021 to December 2021), five virtual meetings, each lasting 60 to 90 min, were held with UK dietitians who were members of the BIMDG-DG. All worked with one or more types of HTs from both adult and paediatric IMD centres. Alkaptonuria was excluded as dietary care was directed from one UK National centre. To reach agreement about a national dietary system for HTs, the Delphi methodology was used [17]. This systemised communication process was chaired by a facilitator. Structured statements on the dietary treatment of HTs were proposed, and dietitians were invited to agree or contest these statements until consensus was achieved. If two thirds of respondents (67%) agreed, this represented consensus; if this was not achieved, another round of statements with supporting evidence were formulated and discussed. Each dietetic team from either a paediatric or adult service was allocated one vote. All centres were given a minimum of 14 days to consider the evidence and vote on each statement.

Statements were created on the following parameters:The general dietary principles to be used for HTI, HTII and HTIII.The dietary exchange system to be adopted: either a 1 g protein exchange system or a combination of tyrosine/phenylalanine (mg/100 g protein). This distinguished between foods that contain 5% phenylalanine and 3–4% tyrosine for each gram of protein from milk and cereals and foods that contain 3–4% phenylalanine and 2–3% tyrosine for each gram of protein from fruit and vegetables [18].Clear definitions for measuring/calculating protein or phenylalanine/tyrosine from manufactured foods and when these can be given as part of the dietary exchange system or given without measurement.

The exchange system for fruit and vegetables received additional attention, with data collected on their protein, phenylalanine and tyrosine content from five nutritional databases:(1)‘McCance and Widdowson’s The Composition of Foods’ 1980, First supplementary amino acid mg/100 g foods [18];(2)National society for phenylketonuria (NSPKU) database [19] and (personal communication);(3)Amino acid composition of food products used in the treatment of patients with disorders of amino acid and protein metabolism [20];(4)United States of America Department of Agriculture (USDA: United States Department of Agriculture, Agriculture Research Service www.usda.gov (accessed on 3 March 2021)) [21];(5)Mevalia website database (www.Mevalia.com (accessed on 3 March 2020)) based on Frida.fooddata.dk version 4 National Food Institute, Technical University of Denmark [22].

Fruits and vegetables were categorised into four groups depending on the protein content per 100 g: protein content ≤1 g/100 g, 1.1 to ≤2 g/100 g, 2.1 to ≤3 g/100 g and 3.1 to ≤4 g/100 g (Appendix A). Tyrosine and phenylalanine analysis data were added as tyrosine mg/100 g and phenylalanine mg/100 g of food. The percentage of tyrosine to protein and percentage of tyrosine to phenylalanine were calculated. A correlation was made for all fruits and vegetables with a protein content of 0 to ≤4 g/100 g to examine the following relationships: phenylalanine vs. tyrosine (mg/100 g), tyrosine (mg/100 g) vs. protein (g/100 g) and the sum of tyrosine and phenylalanine (mg/100 g) vs. protein (g/100 g).

The following additional information was collected: The number and types of HT patients being managed in each treatment centre;The dietary management practices of each treatment centre, including the type of exchange system used;

Ethical approval was not required as per UK Policy Framework for Health and Social Care Research (www.hra.nhs.uk, accessed on 26 March 2021).

## 3. Results

In the UK, 14 IMD centres, 8 paediatric and 6 adult centres cared for patients with HTs, with a median number of 20 (range 17–21) dietitians participating in each meeting. Table 1 shows the distribution of HTs by type and age, together with the number and range of patients per centre. The type and distribution of HTs varied between the 14 centres. Some centres abstained from voting; reasons for abstentions were: ‘centres had limited experience with one or all types of HT, the amount of tyrosine/phenylalanine from fruit and vegetables was dependent on the portion size eaten so were unable to decide how to vote or centres had HT patients following minimal dietary restrictions.’ Therefore, these centres could not be considered as part of the voting system for statements they chose to abstain from voting on.

Table 2 provides a summary from each centre on the current exchange system (protein or phenylamine/tyrosine) used for fruits, vegetables and manufactured foods. Table 3 gives an overview of the Delphi statements and the number of voting rounds required to reach consensus.

During the first round of voting, most centres (100% for HTI, 78% for HTII and 83% HTIII) agreed to use 1 g protein exchanges to calculate/measure all protein-containing foods except for fruit and vegetables. It was also agreed to adopt the same protein cut-off points as the UK PKU dietary guidelines to determine when a protein-containing food should be calculated as an exchange food or given without measurement [16]. The plant milks required three rounds of discussion. Although it was agreed to calculate any plant milk that contained protein >0.1 g/100 mL, it was recognised that the quantity of milk consumption may be less for adults compared to children and this guidance may need to be individualised.

The dietary system for calculating fruit and vegetables received two rounds of Delphi consultations. A correlation was made comparing the following: (1)Tyrosine vs. phenylalanine (mg/100 g for fruit and vegetables) (Figure 1a,b);(2)Tyrosine (mg/100 g for fruit and vegetables) vs. protein (g/100 g) (Figure 1c,d);(3)Protein (g/100 g) vs. the sum of tyrosine and phenylalanine (mg/100 g for fruits and vegetables) (Figure 1e,f).

**Figure 1 nutrients-14-05202-f001:**
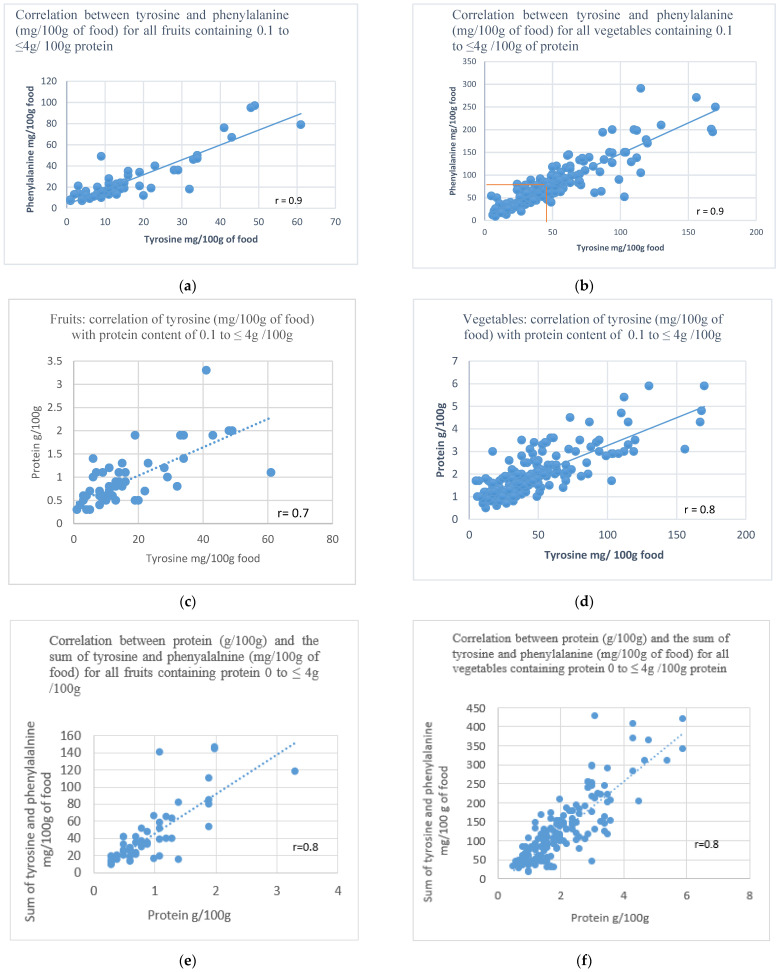
**Correlation between tyrosine and phenylalanine for fruits and vegetables.** (**a**) Correlation between tyrosine and phenylalanine (mg/100 g) for fruits containing 0.1 to ≤4 g/100 g of protein. (**b**) Correlation between tyrosine and phenylalanine (mg/100 g) for vegetables containing 0.1 to ≤4 g/100 g of protein. **Correlation between tyrosine and protein for fruits and vegetables.** (**c**) Correlation between tyrosine (mg/100 g for fruits) and protein (g/100 g) for all fruits containing protein 0.1 to ≤4 g/100 g. (**d**) Correlation between tyrosine (mg/100 g for vegetables) and protein (g/100 g) for all vegetables containing protein 0.1 to ≤4 g/100 g. **Correlation between protein and the sum of tyrosine and phenylalanine for fruits and vegetables.** (**e**) Correlation between protein (g/100 g) and the sum of tyrosine and phenylalanine (mg/100 g) for all fruits containing protein 0.1 to ≤4 g/100 g of fruits. (**f**) Correlation between protein (g/100 g) and the sum of tyrosine and phenylalanine (mg/100 g) for all vegetables containing protein 0.1 to ≤4 g/100 g of vegetables.

These results are shown in Figure 1a–f below.

There was a clear and close correlation (r = 0.9) between tyrosine and phenylalanine (mg/100 g of food) for all fruits and vegetables. Similarly, there was a good correlation with tyrosine (mg/100 g for fruit/vegetables) when compared with protein (g/100 g for fruit/vegetables): r = 0.7 for fruits and r = 0.8 for vegetables. The sum of tyrosine and phenylalanine combined compared with protein g/100 g gave a correlation r = 0.8. A fruit and vegetable containing a phenylalanine content of 75 mg/100 g (the same as the upper cut-off value for calculating/measuring fruits/vegetables in a phenylalanine-restricted diet in PKU [23], correlated to approximately a tyrosine content of 40 mg/100 g for fruits/vegetables (equivalent to a protein content of 1.8 g/100 g). Using the analysis of McCance and Widdowson et al. for foods [24] (excluding fruit and vegetables), the amount of tyrosine provided by 1 g of protein is around 40 mg (4%) compared to 50 mg (5%) for phenylalanine (Table 4). Fruit and vegetables contain a lower percentage of phenylalanine and tyrosine, hence the justification to calculate these differently from other protein-containing foods.

Table 4 describes the higher content of phenylalanine and tyrosine in meat, milk, egg and cereal products; median percentage of phenylalanine is 5% and tyrosine 4% per 1 g of protein. Vegetables are lower ranging from 1.5 to 3.5% for phenylalanine and 1.1 to 3.5% for tyrosine per 1 g of protein [18]. Mushrooms contain a higher percentage of nitrogen from urea giving them a higher protein content.

Across the range of protein values (0 to ≤4 g protein/100 g) for fruits and vegetables, the amount of phenylalanine was greater than the tyrosine content except for a single analysis of jackfruit from the USA, although this differed from the recent UK analysis (NSPKU 2022).

Nine centres voted on the fruit and vegetable exchange system and agreed the same dietary principles could be used for fruit and vegetables in HTs as in PKU [16]. Two centres suggested using protein exchanges, and *n* = 3 centres abstained from voting.

Five individual fruits and vegetables had variable protein and tyrosine/phenylalanine analysis, and their allocations as part of the HT dietary system are summarised in Table 5.

## 4. Discussion

This is the first publication reporting a systematic approach to standardise UK practices for the dietary management of HTs. The aim of dietary treatment is to limit tyrosine and phenylalanine intake while maintaining blood tyrosine and phenylalanine concentrations within target treatment recommendations [1,25]. The majority of authors agree that optimal long-term concentrations of tyrosine are unclear. A review of publications between 2008 and 2020 representing clinicians from different countries recommends that tyrosine concentrations should be maintained ≤500 μmol/L [9,12,13,26,27,28,29,30]. Tyrosine concentrations ≤400 μmol/L are considered safe, with the more recent publications suggesting a plasma tyrosine of ≤400 μmol/L [9,13,28,29,30]. However, recommendations vary between metabolic centres, with some advocating an upper limit of 800 μmol/L [28].

Any dietary practices adopted for HTs must be consistent, logical, effective, straightforward and easy to implement for families and health professionals. No dietary management should be unnecessarily restrictive. The aim of this Delphi process was to agree consensus statements on dietary management using both scientific evidence and practical experience permitting harmonisation of dietary HT practice across all UK centres. All UK IMD dietetic teams caring for HT patients were invited to contribute and all centres participated in at least one Delphi round, with a majority agreeing with the statements.

Several initiatives, issues and supporting data prompted this consultation process. Firstly, the success of the BIMDG dietetic collaborative process in defining consensus statements in PKU. This resulted in consistent and clear guidance on calculating and defining protein exchanges for fruits, vegetables and manufactured foods for PKU, hence a recognised need to standardise the dietary approach for other amino acid conditions [16]. The Delphi methodology actively engaged dietitians, promoting active and open debate and eventual agreement. A facilitator encouraged each dietetic team to contribute their evidence and experience, so it was inclusive of every team. Secondly, in HTI, there is growing evidence that natural protein intake may be higher in some patients without compromising target treatment blood phenylalanine and tyrosine concentrations [8,31]. Work by Bärhold [8] suggests that most vegetables together with other moderate protein foods, for example, cream cheese, normal bread, pasta, pastries, nuts, legumes and eggs, in processed foods could be given as occasional consumption without measurement, and this challenges the stringency of current dietary management. Yilmaz 2020 [31] similarly showed that natural protein tolerance increased with age in a group of 20 children with HTI, and the evidence suggested that a new dietary approach may be necessary when daily protein allowance exceeded 20 g/day. The third reason for engaging in this process was the ambiguity in existing practice. There was discrepancy between the type of fruits and vegetables that were allowed in the HT diet between centres.

For all foods except fruit and vegetables, it was agreed to use the protein analysis (1 g protein exchange system) to determine their suitability in the diet. Work by Evans et al. [15,16] in collaboration with the BIMDG-DG has extensively examined the protein cut-off definitions for different food groups in the PKU dietary system. Deciding protein cut-off points for different foods was a difficult process. Not all food groups/types can be considered in the same way, and cut-off values depend on the portion size of food likely to be consumed and the role of each food in the diet. Having demonstrated the correlation of phenylalanine vs. tyrosine for fruit and vegetables was high (r = 0.9), it seemed logical to adopt the same cut-off values used in PKU [23] which will help rationalise dietary advice for both conditions simultaneously in the future.

The protein, phenylalanine and tyrosine analysis of 171 vegetables and 20 fruits was considered. The median number of amino acid analyses for each vegetable was three (range 1–7) and two (range 1–6) for fruits. There was a total of 21 (11%) ‘one only’ amino acid analyses, which is a recognised limitation. There were some anomalies in the amino acid analysis for the same vegetables and fruits; for example, watercress analysed by USDA (United States Department of Agriculture, Agriculture Research Service) [21], NSPKU and National Food Institute, Technical University of Denmark [22], had three different protein/tyrosine contents per 100 g weight of food: 2.3 g/114 mg, 3 g/26 mg and 1.7 g/46 mg (although the percentage of tyrosine to phenylalanine remains at approximately 55% for all three evaluations).

Differentiating between the lower amounts of tyrosine in vegetables and fruit compared to animal, dairy and cereal products is important. Using exchanges adapted from protein will provide only approximate estimations of tyrosine intake and possibly lead to less dietary freedom. In PKU, many vegetables only contain 35 mg phenylalanine per gram of protein although inconsistencies are recognised [32]. In the UK, current recommendations for calculating fruit and vegetables in HT are variable and conflicting with some centres using 1 g protein exchanges to calculate potatoes and bananas. The near-perfect correlation (r = 0.9) between phenylalanine and tyrosine analysis (mg/100 g of protein) across the range of fruit and vegetables analysed allows the same calculation system for PKU to be used for HTs. By adopting the same system, fruits and vegetables containing a phenylalanine content of ≤75 mg/100 g (or tyrosine ≤40 mg/100 g) can be included in the diet without measurement. Those containing more than 75 mg/100 g (or tyrosine 40 mg/100 g) will be calculated as recommended by the PKU European guidelines 2017 [33].

Amino acid analysis is complex [34,35]; several steps are needed in the analysis process (hydrolysis, separation, detection), and each step requires specific conditions and presents analytical challenges. Standard hydrolysis conditions are not suitable for the extraction of all amino acids, and there is currently no official standardised method for amino acid analysis. The Association of Analytical Communities (AOAC) has validated methods for some but not all amino acids. Precision techniques have improved over time, and accredited laboratories should be chosen to ensure reliable analysis. Analysis is expensive, and repeated analysis is needed for consistency and reproducibility but may be unaffordable. The species of fruit and vegetable, its condition (cooked or raw), its maturity at the time of analysis and the environmental conditions soil, seasonality and climate may alter the amino acid profile particularly when comparing analyses from different countries [36].

The Delphi system allowed parity, and IMD centres who cared for HT patients collaborated in the consultation period and the final statements (Table 6). However, this process was also subjective. There were abstentions in the voting process, which likely reflects the difficulties in making judgments related in part to a limited experience of managing patients with HTs, and some patients may not require dietary intervention as in HTIII. Physicians were not involved in this process as UK dietary practice of HTs is dietetic led. However, the new system chosen advantageously aligns dietary practices between PKU and HTs. Accepting the same dietary system will support sharing of educational resources across the two conditions.

## 5. Conclusions

This collaborative process was constructive, allowing IMD dietitians to focus on the anomalies of current dietary practice in HTs. Consensus was agreed to use 1 g protein exchanges for all dairy and cereal-based foods. For fruits and vegetables, the same dietary principles as the UK PKU dietary guidelines were adopted, basing calculations on their phenylalanine/tyrosine content, which contains a lower percentage concentration per gram of protein compared to animal, milk and cereals. Guidelines help health care professionals to deliver a consistent message. Discussion and consultation produced agreement amongst experienced IMD dietitians to change dietary practices in the management of HTs.

## Figures and Tables

**Table 1 nutrients-14-05202-t001:** Number of patients with tyrosinaemia (HTI, HTII, HTIII) by age, category (paediatric/adult) and number of patients (range) per centre.

Number of Paediatric Centres Caring for HT Patients Aged ≤16 Years (Range for Number of Patients/Centre)	Number of Patients Aged ≤16 Years *n*	Number of Adult Centres Caring for HT Patients Aged ≥17 Years (Range for Number of Patients/Centre)	Number of Patients Aged ≥17 Years *n*	Total Number of Patients *n*
HTI *n* = 8/8 (1–12)	44	HTI *n* = 4/6 (3–12)	31	75
HTII *n* = 5/8 (1)	5	HTII *n* = 4/6 (1–2)	5	10
HTIII *n* = 3/8 (1–4)	7	HTIII *n* = 2/6 (2–3)	7	14

**Table 2 nutrients-14-05202-t002:** Existing practices of each centre on the type of exchange system used (1 g protein exchanges/phenylalanine/tyrosine exchanges) for manufactured foods, fruit and vegetables.

Adult and Paediatric Responses	HTI (*n* = 12 Centres)	HTII (*n* = 9 Centres)	HTII (*n* = 6 Centres)
		**Manufactured Foods**	
Use of 1 g protein exchanges to calculate exchanges from manufactured foods	12/12 (100%)	7/9 (78%)	5/6 (83%)
Use of phenylalanine/tyrosine exchanges	0/12 (0%)	0/9 (0%)	0/6 (0%)
		**Fruit and vegetables**	
The upper protein cut off point (g/100 g) that is used to define when **vegetables** are calculated/measured within the exchange system *			
≤1.0 g	7/8 (88%)	3/5 (60%)	2/3 (67%)
≤1.5 g	1/8 (13%)	1/5(20%)	1/3 (33%)
≤2.0 g	0/8 (0%)	1/5 (20%)	0/3 (0%)
The upper protein cut off point (g/100 g) that is used to define when **fruits** are calculated/measured within the exchange system *			
≤1.0 g	8/9 (89%)	4/6 (67%)	2/3 (67%)
≤1.5 g	1/9 (11%)	1/6 (17%)	1/3 (33%)
≤2.0 g	0/9 (0%)	1/6 (17%)	0/3 (0%)

* some centres abstained from voting and are omitted from the results.

**Table 3 nutrients-14-05202-t003:** Delphi statements (*n* = 11) with the number of voting rounds needed to reach consensus on protein cut-off values and the dietary calculation system used when counting fruit and vegetables in the dietary treatment of HTs.

	Voting Agreement by Treatment Centres			Number or Voting Rounds
	HTI Number of Centres *n* = 12	HTII Number of Centres *n* = 9	HTIII Number of Centres *n* = 6	
**Statement 1: the following protein cut-off point is used to define an exchange-free food i.e., if the protein content exceeds this amount, it should be calculated as an exchange food (this is the same as the UK PKU dietary guidelines)** [16] *****				
An exchange-free food defined as a food not calculated/measured, and when the protein content is ≤0.5 g/100 g (except fruit, vegetables and some manufactured foods e.g., sweets, gravies and desserts)	11/12 (92%)	6/6 (100%)	4/4 (100%)	Round 1
**Statement 2–9: the following manufactured foods should be calculated/measured as part of the protein exchange system if the protein content exceeds the following upper protein amounts given below (this is the same as the UK PKU dietary guidelines)** [16] *****				
Tabletop sauces (e.g., ketchup, brown, chilli, BBQ sauces) containing exchange ingredients with a protein content >1 g/100 g	10/12 (83%)	4/6 (67%)	2/3 (67%)	Round 1
Mayonnaise/salad cream dressings containing exchange ingredients with a protein content >1 g/100 g	11/12 (92%)	5/6 (84%)	2/3 (67%)	Round 1
Cook-in liquid sauces containing exchange ingredients with a protein content >1 g/100 g	12/12 (100%)	6/6 (100%)	3/3 (100%)	Round 1
Soya sauce with a protein content >1.5 g/100 g	10/12 (83%)	4/6 (67%)	2/3 (67%)	Round 1
Special low protein foods containing exchange ingredients with a protein content >0.5 g/100 g	10/11 (91%)	6/6 (100%)	3/3 (100%)	Round 1
Plant milks/special low protein milks are exchange-free if protein content is ≤0.1 g/100 mL; and should be calculated/ measured as part of protein exchange system if the protein content is >0.1 g/100 mL	7/10 (70%)	3/5 (60%)	1/2 (50%)	Round 1
The protein cut off point for plant milks should be: ≤0.1 g/100 mL = exchange-free; >0.1 g/100 mL is an exchange food. This guidance applies to HTI, HTII, HTIII	6/11 (55%)			Round 2
The majority of plant milks should be calculated as a protein exchange in tyrosinemia. However, any plant milk containing a protein content of only 0.1 g/100 mL can be given as exchange-free. This statement applies to HTI, HTII, HTIII	10/11 (91%)			Final Round 3
**Statement 10: Phenylalanine/tyrosine analysis should be used in the allocation of fruit and vegetables in the dietary treatment of HT’s**				
Phenylalanine/tyrosine analysis	9/11 (82%)	5/6 (83%)	3/3 (100%)	Round 2
Protein analysis	2/11 (18%)	1/6 (17%)	0/3 (0%)	
**Statement 11: The same dietary system should be used for Tyrosinaemia Type I, II and III**				
Agreement	9/9 (100%)			Round 2

Legend: HTI, II, III, tyrosinaemia type I, II and III; LP, low protein; PKU, phenylketonuria, * some centres abstained from voting as they had limited experience with HT patients or had patients on less restricted diets associated with poor adherence.

**Table 4 nutrients-14-05202-t004:** The amount of protein (g), phenylalanine (mg) and tyrosine (mg) per 100 g of food and the percentage per gram of protein (analysis taken from Paul et al. 1980 amino acid analysis data) [24].

Food	Protein /100 g	Phe mg /100 g	Tyr mg /100 g	* Phe g /100 g	* Tyr g /100 g	% Phe/g Protein	% Tyr/g Protein
Protein, phenylalanine and tyrosine content of meat, milk, egg							
Beef cooked	29.2	1310	1120	1.3	1.1	4	4
Egg boiled	12.3	630	490	0.6	0.5	5	4
Yoghurt	4.8	280	240	0.3	0.2	6	5
Milk	3.3	180	150	0.2	0.2	5	5
Protein, phenylalanine and tyrosine content of cereal-based foods							
Oats	12.4	660	450	0.7	0.5	5	4
Cornflakes	8.6	430	330	0.4	0.3	5	4
White flour	9.8	520	280	0.5	0.3	5	3
Rice boiled	2.2	110	93	0.1	0.1	5	4
Porridge	1.4	74	50	0.1	0.1	5	4
Protein, phenylalanine and tyrosine content of vegetables							
Mushroom fried	2.2	120	110	0.1	0.1	5	5
Beetroot	1.3	46	46	0.05	0.05	3.5	3.5
Carrots boiled	0.6	17	14	0.02	0.01	3	2
Tomatoes	0.9	15	11	0.02	0.01	1.6	1.2
Turnips	0.9	14	10	0.01	0.01	1.5	1.1

* Figures have been rounded up for cereals, meat, eggs and milk products, Phe—phenylalanine, Tyr—tyrosine; ‘*McCance and Widdowson’s The Composition of Foods*’ 1980 First supplementary amino acid mg/100 g foods (McCance, Widdowson, Paul, Southgate, Russell, Great Britain Medical Research Council, 4th revised and extended edition).

**Table 5 nutrients-14-05202-t005:** Fruit and vegetables with protein and tyrosine/phenylalanine content that crossed over between exchange-free and exchange (calculated/measured) food.

Fruit/Vegetable	Comment	Decision of the Group
Cauliflower	Of 8 different international analyses, 5/8 indicated that cauliflower was low in protein, tyrosine and phenylalanine, and 3/8 suggested it should be considered an exchange vegetable. Further analysis is necessary.	To include as an exchange vegetable until further analysis is available, in line with current recommendations for PKU.
Mushrooms	Seven of 8 analyses suggested a low tyrosine/phenylalanine content so should be considered an exchange-free vegetable, although the protein content was >2 g/100 g for 7/8 analyses. However, mushrooms have a measurable amount of non-protein nitrogen in the form of urea, purines and pyrimidines.	To include as an exchange-free vegetable.
Watercress	There was limited protein and tyrosine/phenylalanine analyses (*n* = 3). The UK analysis suggested that watercress was low in tyrosine/phenylalanine although protein content >2 g/100 g.	To include as an exchange-free vegetable until further analysis is available, in line with current recommendations for PKU.
Avocado	Although the protein content was ≤2.0 g/100 g, 2/6 international phenylalanine analyses exceeded 75 mg/100 g, but 4/6 were ≤75 mg/100 g.	To include as an exchange-free vegetable until further analysis is available, in line with current recommendations for PKU.
Prunes	There was limited protein and tyrosine/phenylalanine analysis (*n* = 1). Although the protein content was >2.0 g/100 g, the tyrosine/phenylalanine content was low.	To include as an exchange-free fruit until further analysis is available.

**Table 6 nutrients-14-05202-t006:** Summary of final consensus statements.

The same dietary principles will be used for all HTs: HTI, II and III. Exchange fruit and vegetables are allocated as part of the protein exchange system according to their tyrosine/phenylalanine content per 100 g of product (tyrosine >40 mg/100 g of product, equivalent to phenylalanine >75 mg/100 g of product). Foods containing exchange ingredients will be calculated/measured using 1 g protein exchanges. Exceptions to this rule follow the same dietary principles as referenced by Evans et al. [15,16] for PKU. Defining upper protein cut-off values and when to count products as an exchange food, it was agreed to use the same guidance as for UK PKU dietary guidelines, based on the evidence from previous detailed and collaborative work with the BIMDG dietitians group. An exchange-free food is defined as a food not calculated/measured when the protein content is ≤0.5 g/100 g (except fruit, vegetables and some manufactured foods, e.g., sweets, gravies and desserts). The following manufactured foods should be calculated as part of the protein exchange system if the protein content exceeds the upper protein amounts given below (this is the same as the UK PKU dietary guidelines) (Evans 2020) [16]: Tabletop sauces (e.g., ketchup, brown, chilli, BBQ sauces) containing exchange ingredients with a protein content >1 g/100 g. Mayonnaise/salad cream dressings containing exchange ingredients with a protein content >1 g/100 g. Cook-in liquid sauces containing exchange ingredients with a protein content >1 g/100 g. Soya sauce with a protein content >1.5 g/100 g. Special low-protein foods containing exchange ingredients with a protein content >0.5 g/100 g. The majority of plant milks should be calculated as a protein exchange in tyrosinaemia. However, any plant milk containing a protein content of only <0.1 g/100 mL can be counted as exchange-free. This statement applies to HTI, HTII and HTIII.

## Data Availability

Not applicable.

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
