# Peer review of "UK Dietary Practices for Tyrosinaemias: Time for Change"

_nutrients, 2022, doi:10.3390/nu14245202_

Round 1

Reviewer 1 Report

I found the article to be an excellent contribution to the follow-up of patients with type 1 Tyrosinemia. It is essential to bear in mind that one of the great problems in follow-up is determining how much tyrosine is being provided by the diet, thinking that it comes from the metabolism of phenylalanine.
That is why within the methodology various sources of food tyrosine content have been evaluated, and a median of this content has been determined. In addition, foods of vegetable origin were left out due to inconsistency in the content of proteins in different references, it seemed to me a very thorough job.
Together with the fact that 14 centers have participated in this evaluation and that they held constant discussions in order to reach agreements, it shows collaborative work based on experience in the follow-up of patients with Tyrosinemia type I, II, and III.
The contribution provided by the article is of great value for clinicians who carry out the frequent follow-up of patients with Tyrosinemia, especially patients with Tyrosinemia type I, which is the most complex.
Facilitating monitoring through food exchange, considering protein intake and its tyrosine equivalence, is a tool that enables nutritional planning for nutritionists, but especially for families.
It is for this reason that I found the remarkable work and a great contribution to nutrition in patients with Tyrosinemia.

Author Response

Thank you for the helpful comments please see the letter attatched

Reviewer 2 Report

This manuscript reports the consensus of dietitians in the UK on dietary management of various forms of tyrosinemia. It needs to be mentioned that the outcome does not necessarily reflect the opinion of expert physicians caring for patients with tyrosinemia in the UK, though. Furthermore, it is imperative to highlight that the treatment target for tyrosine - stated as 200-400 micromole/L - is not supported by evidence. Rather it is using a publication that didn’t even make it to the print version of JIMD (van Dam et al., JIMD Rep). In fact, expert opinion dictates to maintain tyrosine below 400-500 micromole/L (De Laet et al. 2013, Orphanet J Rare Dis). Please note, that is a distinct difference to the 200-400 micromole/L target. The treatment target stated in the current manuscript may cause overtreatment and potentially leading to harm that would be iatrogenic of nature. It is also prudent to mention in the manuscript that even formal data to support the expert recommendation of 400-500 are lacking.  The current evidence and rationale for treatment targets needs to be properly discussed. Why is it that we want to reduce tyrosine in NTBC treated HT1 patients? The answer is to avoid eye lesions. We know from HT2 patients that eye and skin lesions are rarely seen with plasma tyrosine <800, suggesting that levels should be maintained below this level. However, the repeated observation of developmental delay in HT2 patients and some HT3 patients suggests that a lower level, perhaps 400-500 micromole/L, may be appropriate.

Author Response

Thank you for your helpful comments. Please see letter attached

Round 2

Reviewer 2 Report

The comment inserted - 'long term studies are urgently needed to determine the optimal target concentrations for blood tyrosine' - is not sufficient. To avoid potential overtreatment of patients readers need to understand and the authors must clearly state that the tyrosine targets applied in their review ARE NOT EVIDENCE BASED, and that LEADERS IN THE FIELD CONSIDER HIGHER TYROSINE TARGETS AS SAVE. 

Author Response

Dear Reviewer 2

Thank you for your comments. Please see the full response to your comments in the attached letter. We have also amended the manuscript accordingly.

Thank you
